# Household transmission of SARS-CoV-2 Omicron variant of concern subvariants BA.1 and BA.2 in Denmark

Frederik Plesner Lyngse [1,2,3] ✉, Carsten Thure Kirkeby[4], Matthew Denwood [4], Lasse Engbo Christiansen [5], Kåre Mølbak [3,4], Camilla Holten Møller[3], Robert Leo Skov [3], Tyra Grove Krause[3], Morten Rasmussen [3], Raphael Niklaus Sieber [3], Thor Bech Johannesen[3], Troels Lillebaek[3,6], Jannik Fonager [3], Anders Fomsgaard [3], Frederik Trier Møller [3], Marc Stegger [3], Maria Overvad[3], Katja Spiess [3] & Laust Hvas Mortensen [7,8]

SARS coronavirus 2 (SARS-CoV-2) continues to evolve and new variants emerge. Using nationwide Danish data, we estimate the transmission dynamics of SARS-CoV-2 Omicron subvariants BA.1 and BA.2 within households. Among 22,678 primary cases, we identified 17,319 secondary infections among 50,588 household contacts during a 1–7 day follow-up. The secondary attack rate (SAR) was 29% and 39% in households infected with Omicron BA.1 and BA.2, respectively. BA.2 was associated with increased susceptibility of infection for unvaccinated household contacts (Odds Ratio (OR) 1.99; 95%-CI 1.72-2.31), fully vaccinated contacts (OR 2.26; 95%-CI 1.95–2.62) and booster-vaccinated contacts (OR 2.65; 95%-CI 2.29–3.08), compared to BA.1. We also found increased infectiousness from unvaccinated primary cases infected with BA.2 compared to BA.1 (OR 2.47; 95%-CI 2.15–2.84), but not for fully vaccinated (OR 0.66; 95%-CI 0.57–0.78) or booster-vaccinated primary cases (OR 0.69; 95%-CI 0.59–0.82). Omicron BA.2 is inherently more transmissible than BA.1. Its immune-evasive properties also reduce the protective effect of vaccination against infection, but do not increase infectiousness of breakthrough infections from vaccinated individuals.

The current pandemic caused by SARS-CoV-2 is characterized by continuous emergence of new variants taking over from previous variants as a result of natural selection[1]. Most recently, the Omicron variant of concern (VOC), Pango lineage B.1.1.529, has become the most prevalent in most countries in Europe as well as the rest of the world[2]. Of the previously identified Omicron subvariants in early 2022[3,4], three subvariants had been detected in Denmark, namely BA.1.1, BA.1 and BA.2, where the latter two by far have been the most

abundant. BA.1 and BA.2 differ by approximately 40 mutations[5] in addition to a key deletion at position 69-70 in the spike region of BA.1 compared to BA.2[6,7].

BA.1 was first detected in Denmark on 25th November 2021, and BA.2 was first detected on 5th December 2021. Since then, the prevalence of BA.2 has been increasing faster than that of BA.1. In week 52 of 2021, BA.2 accounted for around 20% of all Danish SARS-CoV-2 cases. Two weeks later the proportion had increased to around 45%,

[1]Department of Economics & Center for Economic Behavior and Inequality, University of Copenhagen, Copenhagen, Denmark. [2]Danish Ministry of Health, Copenhagen, Denmark. [3]Statens Serum Institut, Copenhagen, Denmark. [4]Department of Veterinary and Animal Sciences, Faculty of Health and Medical Sciences, University of Copenhagen, Copenhagen, Denmark. [5]Department of Applied Mathematics and Computer Science; Dynamical Systems, Technical University of Denmark, Kgs. Lyngby, Denmark. [6]Global Health Section, University of Copenhagen, Copenhagen, Denmark. [7]Statistics Denmark, Copenhagen, Denmark. [8]Department of Public Health, Faculty of Health and Medical Sciences, University of Copenhagen, Copenhagen, Denmark. ✉e-mail: fpl@econ.ku.dk

indicating that BA.2 carries an advantage over BA.1 within the highly vaccinated population of Denmark. The RT-PCR test used in Denmark does not target the S-gene deletion to detect Omicron cases but instead targets the spike position L452 Wt[8]. Thus, in the current set-up, Danish RT-PCR data cannot distinguish between BA.1 and BA.2. However, whole genome sequencing (WGS) is conducted routinely in Denmark (www.covid19genomics.dk), providing the opportunity to identify and differentiate between BA.1 and BA.2.

We have previously used a model of household transmission to quantify the transmissibility of VOCs, and applied this model to show that the Omicron VOC had an advantage over the Delta VOC due to immune evasiveness[9].

The increasing numbers of BA.2 cases justify the questions we address in this study; (1) Is there a difference in the household transmission patterns between Omicron VOC subvariant BA.1 and BA.2; and (2) if there is a difference, is it due to a difference in susceptibility, infectiousness, or both, and could this indicate a difference in immune evasiveness between the two subvariants?

## Results

We identified 11,348 households with Omicron BA.2 comprising a total of 25,859 household contacts, of which 10,102 tested positive within 7 days, resulting in a secondary attack rate (SAR) of 39% (Table 1). Similarly, we identified 11,330 households with Omicron BA.1 comprising a total of 24,729 household contacts, of which 7217 tested positive, resulting in a SAR of 29%. The distributions of age, sex, household size

and vaccination status of primary cases and household contacts were broadly comparable between BA.1 and BA.2 households (Table 1).

The cumulative probability of contacts being tested at least once increased from 37% to 84% for Omicron BA.1 contacts (blue) and from 37% to 83% for BA.2 contacts (red) at 7 days after the primary case tested positive (Fig. 1a). The cumulative probability of contacts being tested at least twice increased from 9% to 61% for BA.1 contacts and from 8% to 58% for BA.2 contacts 7 days after the primary case tested positive. In households infected with the Omicron BA.2 (red), the SAR was 6% on day 1 and 39% on day 7 (Fig. 1b). Similarly, in households infected with BA.1 (blue), the SAR was 6% and 29%, respectively. (See Fig. S2 for a 14 day follow-up.)

The effect of vaccination on susceptibility to infection and infectiousness of SARS-COV-2 within households is presented in Table 2. We stratified the effects of susceptibility and infectiousness by variant because we observed an interaction between variant and vaccination status of the contacts ($p < 0.0001$) and between variant and vaccination status of the primary case ($p = 0.0048$).

For households infected with Omicron BA.2, we estimated an OR of susceptibility to infection of 1.12 (95%–CI 1.03–1.22) for unvaccinated contacts and an OR of 0.81 (95%–CI 0.75–0.87) for booster-vaccinated contacts, both compared to fully vaccinated contacts, after adjustment for confounders (age and sex of the contact, age and sex of the primary case, household size, and primary case sample date) (Table 2). The corresponding OR estimates of susceptibility for households infected with Omicron BA.1 were 1.27 (95%–CI 1.17–1.39)

## Table 1 | Summary Statistics (primary cases and contacts reported separately)

| | Omicron - BA.2 | | | | Omicron - BA.1 | | | |
|---|---|---|---|---|---|---|---|---|
| | Primary cases | Household contacts | Secondary cases | SAR (%) | Primary cases | Household contacts | Secondary cases | SAR (%) |
| **Total** | 11,348 | 25,859 | 10,102 | 39 | 11,330 | 24,729 | 7217 | 29 |
| **Sex** | | | | | | | | |
| Male | 5504 | 13,040 | 4778 | 37 | 5487 | 12,450 | 3454 | 28 |
| Female | 5844 | 12,819 | 5324 | 42 | 5843 | 12,279 | 3763 | 31 |
| **Age** | | | | | | | | |
| 0–9 years | 2018 | 4922 | 1946 | 40 | 1123 | 4619 | 1457 | 32 |
| 10–19 years | 3287 | 4470 | 1491 | 33 | 2909 | 4513 | 1198 | 27 |
| 20–29 years | 1788 | 2571 | 979 | 38 | 2496 | 3170 | 928 | 29 |
| 30–39 years | 1406 | 4067 | 2209 | 54 | 1646 | 3146 | 1330 | 42 |
| 40–49 years | 1078 | 5588 | 2119 | 38 | 1254 | 4919 | 1307 | 27 |
| 50-59 years | 1015 | 2963 | 918 | 31 | 1082 | 3010 | 645 | 21 |
| 60–69 years | 477 | 864 | 296 | 34 | 520 | 913 | 230 | 25 |
| 70+ years | 279 | 414 | 144 | 35 | 300 | 439 | 122 | 28 |
| **Household size** | | | | | | | | |
| 2 persons | 3675 | 3675 | 1529 | 42 | 4087 | 4087 | 1278 | 31 |
| 3 persons | 2674 | 5348 | 1961 | 37 | 2756 | 5512 | 1491 | 27 |
| 4 persons | 3438 | 10,314 | 4180 | 41 | 3053 | 9159 | 2830 | 31 |
| 5 persons | 1283 | 5132 | 1975 | 38 | 1199 | 4796 | 1329 | 28 |
| 6 persons | 278 | 1390 | 457 | 33 | 235 | 1175 | 289 | 25 |
| **Vaccination status** | | | | | | | | |
| Unvaccinated[a] | 3285 | 6837 | 2839 | 42 | 2497 | 6683 | 2240 | 34 |
| Fully vaccinated[b] | 4667 | 7975 | 3293 | 41 | 5844 | 9458 | 2949 | 31 |
| Booster vaccinated | 3396 | 11,047 | 3970 | 36 | 2989 | 8588 | 2028 | 24 |

[a]Unvaccinated includes individuals with partial vaccination (24 primary cases and 18 contacts). [b]Fully vaccinated includes unvaccinated individuals with previous infection.
Notes: Summary statistics for primary cases are shown separately from summary statistics for household contacts, secondary cases and secondary attack rate (SAR). For example, there were 2018 primary cases aged 0–9 years with Omicron BA.2 and a total of 4922 contacts aged 0–9 years living in households infected with Omicron BA.2. Of the 4922 household contacts, 1946 tested positive, yielding a SAR of 40%. Thus, the SAR reflects the proportion of household contacts that tested positive, irrespective of the characteristics of the primary case. Summary statistics stratified by the primary case level are presented in Tables S4 and S5. The proportions (%) for each category is presented in Table S3.

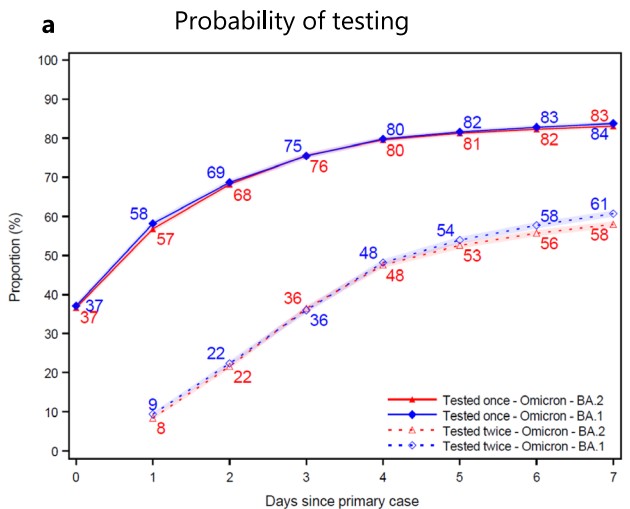

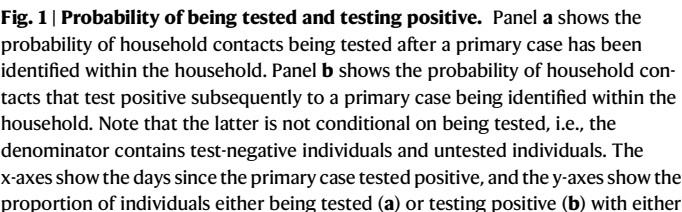

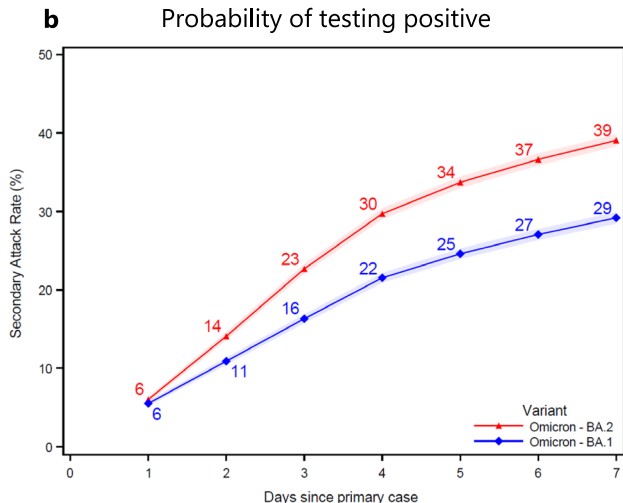

**Fig. 1 | Probability of being tested and testing positive.** Panel **a** shows the probability of household contacts being tested after a primary case has been identified within the household. Panel **b** shows the probability of household contacts that test positive subsequently to a primary case being identified within the household. Note that the latter is not conditional on being tested, i.e., the denominator contains test-negative individuals and untested individuals. The x-axes show the days since the primary case tested positive, and the y-axes show the proportion of individuals either being tested (**a**) or testing positive (**b**) with either

antigen or RT-PCR tests, stratified for the subvariant of the primary case. The SAR for each day according to the subvariant primary case can be read directly from panel **b**. For example, the SAR on day 7 is 39% for BA.2 (red) and 29% for BA.1 (blue), whereas the SAR on day 4 is 30% and 22%, respectively. The markers show the point estimates of the mean. The shaded areas show the 95% confidence bands with cluster-robust standard errors at the household level. Figure S2 presents the two panels with a 14 day follow-up period. Figure S3 presents the 14 day SAR for Omicron BA.1, BA.2, and Delta VOC, as well as those without a known variant.

**Table 2 | Effect of Vaccination**

| | Susceptibility | | Infectiousness | |
|---|---|---|---|---|
| | (Household contacts) | | (Primary case) | |
| | Omicron BA.2 households | Omicron BA.1 households | Omicron BA.2 households | Omicron BA.1 households |
| Unvaccinated | 1.12 | 1.27 | 1.19 | 0.98 |
| | (1.03–1.22) | (1.17–1.39) | (1.08–1.31) | (0.89–1.08) |
| Fully vaccinated | ref | ref | ref | ref |
| | (.) | (.) | (.) | (.) |
| Booster vaccinated | 0.81 | 0.69 | 0.86 | 0.82 |
| | (0.75–0.87) | (0.64–0.75) | (0.78–0.94) | (0.75–0.91) |

Notes: This table shows odds ratio (OR) estimates of susceptibility and infectiousness by vaccination status. Number of observations=50,588; Number of households=22,678. Column 1 shows the susceptibility to infection based on the vaccination status of the household contacts, conditional on living in a household infected with BA.2. Column 2 shows the susceptibility to infection based on the vaccination status of the household contacts, conditional on living in a household infected with BA.1. Column 3 shows the infectiousness based on the vaccination status of the primary case, conditional on living in a household infected with BA.2. Column 4 shows the infectiousness based on the vaccination status of the primary case, conditional on living in a household infected with BA.1. Note that all estimates are from the same model, but with a different reference category across column 1-4. The estimates are adjusted for age and sex of the primary case, age and sex of the household contact, size of the household, and primary case sample date. The estimates are furthermore adjusted for vaccination status of the household contact and primary case interacted with the household subvariant. 95%-confidence intervals are shown in parentheses with cluster-robust standard errors at the household level. The odds ratio estimates for the full model are presented in the appendix Table S12, column I. Figure S5 presents the estimates with different contrasts and reference categories.

for unvaccinated contacts and 0.69 (95%-CI 0.64–0.75) for booster-vaccinated contacts, compared to fully vaccinated contacts.

For households infected with Omicron BA.2, we estimated an OR of infectiousness of 1.19 (95%-CI 1.08–1.31) for unvaccinated primary cases and an OR of 0.86 (95%-CI 0.78–0.94) for booster-vaccinated primary cases, compared to fully vaccinated primary cases. The corresponding OR estimates of infectiousness for households infected with Omicron BA.1 were 0.98 (95%-CI 0.89–1.08) for unvaccinated primary cases and 0.82 (95%-CI 0.75–0.91) for booster-vaccinated primary cases, compared to fully vaccinated primary cases.

Overall, these estimates display a baseline association between vaccination status and both susceptibility and infectiousness.

The relative difference in SAR between the Omicron BA.2 and BA.1 subvariants when comparing across contacts and primary cases with the same vaccination is presented in Table 3. When comparing household contacts with the same vaccination status living in

households infected with Omicron BA.2 relative to living in households infected with Omicron BA.1, we found an increased susceptibility of infection across all vaccination categories. Across unvaccinated contacts, we found an OR of 1.99 (95%-CI 1.72–2.31); across fully vaccinated contacts, we found an OR of 2.26 (95%-CI 1.95–2.62); and, across booster-vaccinated contacts, we found an OR of 2.65 (95%-CI 2.29–3.08).

When comparing primary cases with the same vaccination status living in households infected with Omicron BA.2 relative to those living in households infected with Omicron BA.1, we found an increased infectiousness among unvaccinated primary cases (OR 2.47; 95%-CI 2.15–2.84) and a decreased infectiousness among primary cases that were fully vaccinated (OR 0.66; 95%-CI 0.57–0.78) or booster vaccinated (OR 0.69; 95%-CI 0.59–0.82).

We compared the sample Ct values of primary cases infected with Omicron BA.2 relative to BA.1 and found limited evidence of a

**Table 3 | Relative effect of Omicron VOC BA.2 vs. BA.1**

| | Susceptibility | | | Infectiousness | | |
| --- | --- | --- | --- | --- | --- | --- |
| | (Household contacts) | | | (Primary case) | | |
| | Unvaccinated | Fully vaccinated | Booster vaccinated | Unvaccinated | Fully vaccinated | Booster vaccinated |
| Omicron BA.2 households | 1.99 | 2.26 | 2.65 | 2.47 | 0.66 | 0.69 |
| | (1.72–2.31) | (1.95–2.62) | (2.29–3.08) | (2.15–2.84) | (0.57–0.78) | (0.59–0.82) |
| Omicron BA.1 households | Ref | Ref | Ref | Ref | Ref | Ref |
| | (.) | (.) | (.) | (.) | (.) | (.) |

Notes: This table shows odds ratio (OR) estimates for the relative difference in SAR for households infected with BA.2 compared to BA.1. Number of observations=50,588; Number of households=22,678. Column 1 shows the relative susceptibility to infection, when comparing unvaccinated contacts living in households infected with BA.2 to unvaccinated contacts living in households infected with BA.1. Column 2 shows the relative susceptibility to infection, when comparing fully vaccinated contacts living in households infected with BA.2 to fully vaccinated contacts living in households infected with BA.1. Column 3 shows the relative susceptibility to infection, when comparing booster-vaccinated contacts living in households infected with BA.2 to booster-vaccinated contacts living in households infected with BA.1. Column 4 shows the relative infectiousness, when comparing unvaccinated primary cases living in households infected with BA.2 to unvaccinated primary cases living in households infected with BA.1. Column 5 shows the relative infectiousness, when comparing fully vaccinated primary cases living in households infected with BA.2 to fully vaccinated primary cases living in households infected with BA.1. Column 6 shows the relative infectiousness, when comparing booster-vaccinated primary cases living in households infected with BA.2 to booster-vaccinated primary cases living in households infected with BA.1. Note, all estimates are from the same model, but with a different reference category across column 1-6. The estimates are adjusted for age and sex of the primary case, age and sex of the household contact, size of the household, and primary case sample date. The estimates are furthermore adjusted for vaccination status of the household contact and primary case interacted with the household subvariant. 95%-confidence intervals are shown in parentheses with cluster-robust standard errors at the household level. The odds ratio estimates for the full model are presented in Table S12, column I.

difference in the distribution of sample Ct values (Fig. S4 and Table S6). Adjustment for Ct values of the primary cases did not materially change the findings (Table S15, model XIII). This suggests that the increased transmission of Omicron BA.2 cannot be explained by differences in the viral load of the primary cases.

We allowed for a 14-day follow-up period instead of a 7-day period and found a 14-day SAR of 46% for BA.2 and 36% for BA.1, compared to a 7-day SAR of 29% and 39%, respectively, (Fig. S2). The results only changed slightly when increasing the follow-up period, indicating that our main results are robust to a longer follow-up period (Table S12).

The probability that a positive RT-PCR test was selected for WGS was stable at around 7% across Ct values <35 and age (Fig. S1), suggesting no sampling bias in selection of positive samples for WGS.

We found limited evidence of misclassification of primary and secondary cases distorted our results (appendix section S4.2). (i) We found no evidence of a differential effect of tertiary cases being misclassified as secondary cases across subvariants (Table S10). (ii) Our results were robust to only including households, where all contacts had tested negative after the primary case, indicating that our results were not a result of misclassification of primary cases (Table S13). (iii) We found that 2–7% of the secondary cases were infected with another variant than the primary case, indicating a limited effect of misclassification of secondary cases being infected by the community instead of the household (Table S11).

## Discussion

Our results show that the Omicron BA.2 subvariant is generally more transmissible than the BA.1 subvariant across all groups of sex, age, household size and vaccination group (Table 1 and S4). Both unvaccinated, fully vaccinated and booster-vaccinated household contacts had a higher susceptibility to infection for BA.2 compared to BA.1, indicating an inherent increased transmissibility of BA.2 (Table 3). However, the relative increase in susceptibility was significantly greater in vaccinated contacts compared to unvaccinated contacts (Fig. S5), which points towards immune evasive properties of the BA.2 conferring an even greater advantage for BA.2 in a highly vaccinated population such as Denmark. Unvaccinated primary cases infected with BA.2 had a higher infectiousness compared to those infected with BA.1 (Table 3). Contrary to this, fully vaccinated and booster-vaccinated individuals had a reduced infectiousness. This indicates that after a breakthrough infection, vaccination protects against further transmission, and more so for BA.2 than BA.1. This mechanism is only possible to identify in studies that take the exposure of individuals into account.

Efficient transmission to vaccinated individuals corroborates previous findings that the Omicron VOC possess immune evasive properties[9–12]. However, both booster-vaccinated individuals and fully-vaccinated individuals had reduced susceptibility to infection and infectiousness compared to unvaccinated individuals for both subvariants, suggesting that the effectiveness of vaccines remains significant (Fig. S5). The reduced infectiousness among vaccinated individuals with a breakthrough infection can be explained by a shorter period of viral shedding. This is corroborated by a cohort study from South Korea, where fully vaccinated individuals had a shorter duration of viable viral shedding and a lower secondary attack rate than partially vaccinated or unvaccinated individuals[13]. Studies from China[14,15], and the United States also indicate that vaccination shortens the duration of time of high transmission potential, minimizes symptom duration, and furthermore may restrict tissue dissemination of active virus[16].

Interestingly, our data (Table 3) indicate that the immune escape of BA.2, among vaccinated individuals, mainly leads to an increased susceptibility to infection, whereas its infectiousness is reduced compared with BA.1 reinfections. Hence, it may be incorrect to assume that there is a constant relation between susceptibility and infectiousness in the evolution of SARS-CoV-2 in its attempts to gain foothold in a well vaccinated population. It is beyond the scope of this epidemiological paper to investigate this observation in detail. However, a strain-specific compartmentalization of virus could be a possible biological mechanism, which warrants further studies.

This study has a number of strengths. Firstly, Denmark is, to the best of our knowledge, the only country in the world that has been able to identify a large amount of both BA.1 and BA.2 cases in December 2021 and January 2022. Secondly, any bias introduced in the identification of the subvariants will presumably affect both BA.1 and BA.2 in a similar way. Third, this study draws on exhaustive population registers with a high quality of information covering the whole population. Fourth, in December 2021 and January 2022 Denmark had a high test capacity, around 10% of the Danish population were tested each day and household contacts were frequently being tested: 83–84% at least once and 58–61% at least twice.

Some limitations apply to this study. Our analysis assumes that the timing of positive tests within households can be used to infer primary and secondary infections within the household. It is likely that our study misclassifies a small fraction of secondary cases, but our sensitivity analyses suggest that the potential bias in terms of the comparison across subvariants is limited (appendix section S4.2). The study period runs over Christmas 2021 and New Year's Eve 2021/22, which are public holidays in Denmark. Despite government advice to

limit social activity, it is likely that there has been considerable social mixing with family and friends outside the households during this period. Social mixing over the holidays in conjunction with the high incidence levels in Denmark during this period could in theory mean an increased misclassification due to infections being picked up outside the household. However, this potential bias would be applicable to both subvariants. Moreover, our estimates were robust when excluding households, where the primary case was infected during the holidays (Table S14).

The present household study showed a transmission advantage of Omicron BA.2 over BA.1. Although vaccinations, in particular booster vaccinations, did protect against infection, the 2.26 (fully vaccinated) and 2.65 (booster vaccinated) fold higher odds of infection in BA.2 households indicate that BA.2 as a phenotype represents a further step in immune evasion in the Omicron lineage. However, it is likely that this change came with an evolutionary cost for BA.2. To our surprise, we found a decreased infectiousness of BA.2 relative to BA.1 among fully vaccinated and booster-vaccinated primary cases. Based on such a considerable loss in infectiousness among vaccinated individuals, it is not straightforward to predict the future trajectory of BA.2 relative to BA.1 or other potentially emerging variants.

Evolution of SARS-CoV-2 variants, including the Omicron VOC, is constantly evolving, especially during periods of high transmission in many countries. For public health, it is reassuring that BA.2, like BA.1, seems to be associated with favorable outcomes relative to the Delta variant, and that vaccines protect in particular against hospital admissions and severe illness[17,18]. Even with the emergence of BA.2, vaccines have an effect against infection, transmission and severe disease, although reduced compared to the ancestral variants. The combination of high incidence of a relative innocuous subvariant raised optimism[19]. It is, however, important to follow the future evolution of the SARS-CoV-2 Omicron subvariants closely, as well as future emergent subvariants. Thus, it is critical to maintain rapid high-quality WGS with random sampling as part of surveillance to continuously support the risk assessment of new variants, their impact on public health and to inform public health policymakers, when navigating during a pandemic.

## Methods

### Study design and participants

In this study, we used Danish register data comprising all individuals in Denmark. We linked all individuals to households by their personal identification number. We only included households with 2-6 members to exclude care facilities etc. We linked this with information on all antigen and RT-PCR tests for SARS-CoV-2 from the Danish Microbiology Database (MiBa[20]), and records in the Danish Vaccination Register[21]. We used data on primary cases from 20th December 2021 to 28th January 2022, and allowed a 7-day follow-up period for household contacts, i.e., until 4th February 2022. On 20th December 2021, Omicron BA.2 comprised 5% of all infections, and Omicron BA.1 comprised 60%, while Delta comprised 32%. By 28th January 2022, the proportions were 83%, 16%, and 0%, respectively (Tables S1 and S2).

A primary case was defined as the first individual in a household testing positive with an RT-PCR test within the study period and being identified, by WGS, with the Omicron VOC BA.1 or BA.2. We excluded all households with a positive RT-PCR test within the previous 60 days, which also excludes previous infections with the Omicron VOC. We followed all tests (positive, negative, and inconclusive) of other household members in the follow-up period. A secondary case was defined by either a positive RT-PCR test or a positive antigen test. Households were categorized as BA.1 or BA.2 households depending on the WGS result of the sample from the primary case.

In the study period, a total of approximately 80,000 mid- and high-quality SARS-CoV-2 genomes were produced (Tables S1 and S2). Briefly, sequencing of positive SARS-CoV-2 samples was performed using short

read Illumina technology with the Illumina COVIDSeq Test kit. The library preparation was performed as described by the manufacturer with spike-in of amplicon 64, 70 and 74 from the ARTIC v3 amplicon sequencing panel (https://artic.network). Samples were sequenced on either the NextSeq or NovaSeq platforms (Illumina). Consensus sequences were called using an in-house implementation of IVAR (version 1.3.1) with a custom BCFtools[22] command for consensus calling. The resulting consensus sequences were considered for variant calling when containing <3,000 ambiguous sites including N's. Variants were called using Pangolin (version 04.00.06) on the consensus sequences with pango-designation (version 1.9) or PUSHER (version 1.9) assignment algorithm for known and novel sequences, respectively[23].

The vaccination status of all individuals was classified into three groups following Lyngse et al.[9]: (i) unvaccinated, (ii) fully vaccinated, or (iii) booster vaccinated. Unvaccinated individuals were defined as individuals with no vaccination and no previous infection, and also included 42 partially vaccinated individuals. Fully vaccinated individuals were defined by days since vaccination and the type of vaccine: 7 days after second dose of Comirnaty (Pfizer/BioNTech), 15 days after second dose of Vaxzevria (AstraZeneca), 14 days after second dose of Spikevax (Moderna), 14 days after vaccination of Janssen (Johnson & Johnson), and 14 days after the second dose for cross-vaccinated individuals. Individuals with a previous infection was also defined as fully vaccinated. Individuals were defined as booster vaccinated 7 days after receiving the booster vaccination[24,25]. By 22nd December 2021, 85% of all vaccinated individuals in Denmark were vaccinated with Comirnaty, 14% with Spikevax, 1% with Janssen, and approximately 0% with AstraZeneca[26].

### Statistical analyses

The secondary attack rate (SAR) was defined as the proportion of household contacts within the same household that tested positive between 1–7 days following the positive test of the primary case in that household. We estimated the adjusted odds ratios (OR) of infection in a multivariable logistic regression model. The outcome variable in this model was the binary test result of testing positive or not of each household contact. We used the subvariant as an explanatory variable as well as fixed effects for other potentially confounding variables; age and sex of the primary case, age and sex of the household contact, household size (2-6 members), and primary case sample date to control for time related effects. To test if the subvariants behaved differently depending on the vaccination status of the primary cases (i.e., different infectiousness) and the household contacts (i.e., different susceptibility), we included interactions between household subvariant and vaccination status of the primary cases and the household contacts, respectively. In particular, we estimated the following equation:

$$\log\left(\frac{y_{c,p}}{1-y_{c,p}}\right) = Constant + Variant_p + VaccineStatus_c + VaccineStatus_p$$
$$+ Variant_p \times VaccineStatus_c + Variant_p \times VaccineStatus_p$$
$$+ Age_p + Sex_p + HouseholdSize_p + Age_c + Sex_c + SampleDate_p,$$

$$(1)$$

where $y_{c,p}$ equals one if the contact $c$ is tested positive 1–7 days after exposure to primary case $p$, and zero otherwise. $Variant_p$ determines if the primary case $p$ was infected with Omicron BA.1 or BA.2. $VaccineStatus$ represents the fixed effects of vaccination status (categorical variable) for the primary case $p$ and the contact $c$. Age represents fixed effects of sex, and $HouseholdSize$ represents fixed effects of household size (categorical variable). $SampleDate$ represents fixed effects of the sampling date of the primary cases's positive test (categorical variable). Cluster-robust standard errors were implemented with clustering at the household level by using Taylor series linearization to estimate the covariance matrix of the regression coefficients[27].

In addition, we conducted a series of supplementary analyses to support our main analysis. We provide alternative presentation of our main results (appendix section S3) and measures of model fit (Table S9). We compare a number of alternative specifications of the logistic regression model to assess the robustness of our results (appendix section S4.3). We investigate the effect of using a follow-up period of 14 days instead of 7 days (Fig. S2 and Table S12). Our study relies on the assumption that we correctly identify primary and secondary cases, and that secondary cases are infected by the household primary cases and not from the external community. To examine the potential misclassification of primary and secondary cases, we performed a number of robustness analyses (appendix section S4.2). First, to investigate the potential role of tertiary cases, we compared the SAR across two-person and multi-person households, as there is no tertiary cases in two-person households. Second, to investigate the potential role of misclassification of primary cases, we leveraged the high test capacity in Denmark and restricted our sample to only include households, where all contacts had tested negative after the primary case, in order to rule out contacts without a test and secondary cases that potentially could be the primary case. Third, to investigate the role of secondary cases being infected in the community and not in the household, we estimated the probability of the secondary cases having been infected with the same variant as the primary case. We furthermore, investigated this probability by looking at households located in communities primarily with another variant and households located in communities with a high overall incidence of cases, in order to increase the probability of identifying misclassification.

Our study also relies on unbiased sampling of positive samples for WGS. To investigate if there was a sampling bias, we investigated the sampling probability by age and sample Ct value (Fig. S1).

### Ethical statement

This study was conducted using data from national registers only. According to Danish law, ethics approval is not needed for this type of research. All data management and analyses were carried out on the Danish Health Data Authority's restricted research servers with project number FSEID-00004942. The study only contains aggregated results and no personal data. The study is, therefore, not covered by the European General Data Protection Regulation (GDPR).

### Reporting summary

Further information on research design is available in the Nature Research Reporting Summary linked to this article.

## Data availability

The data used in this study are available under restricted access due to Danish data protection legislation. The data are available for research upon request to The Danish Health Data Authority and Statens Serum Institut and within the framework of the Danish data protection legislation and any required permission from authorities. We performed no data collection or sequencing specifically for this study. Consensus genome data from the Danish cases are routinely shared publicly at GISAID (www.gisaid.org).

## Code availability

The code used for this study can be downloaded from a public repository: https://github.com/Flyngse/SARS-CoV-2_Omicron_BA1_BA2

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

## Acknowledgements

We thank Statens Serum Institut and The Danish Health Data Authority for collecting and providing access to data access. We also thank the rest of the Expert Group for Mathematical Modeling of COVID-19 at Statens Serum Institut for helpful discussions. The authors wish to thank the Danish Covid-19 Genome Consortium for genotyping SARS-CoV-2 positive samples. We thank Simon Kyllebæk Andersen and Carl Benjamin Simpson (Department of Economics, University of Copenhagen) for proofreading the manuscript.

Frederik Plesner Lyngse is supported in part by grants from the Independent Research Fund Denmark (Grant no. 9061-00035B.); Novo Nordisk Foundation (grant no. NNF17OC0026542); the Danish National Research Foundation through its grant (DNRF-134) to the Center for Economic Behavior and Inequality (CEBI) at the University of Copenhagen. Laust Hvas Mortensen is supported in part by grants from the Novo Nordisk Foundation (grant no. NNF17OC0027594, NNF17OC0027812).

## Author contributions

F.P.L. performed all data analyses. M.D. calculated the contrasts between vaccination groups. F.P.L., C.T.K. and L.H.M. wrote the first draft. F.P.L., C.T.K., M.D., L.E.C., K.M., C.H.M., R.L.S., T.G.K., M.R., R.N.S., T.B.J., T.L., J.F., A.F., F.R.M., M.S., M.O., K.S., and L.H.M. contributed to the discussion, revised the first draft and approved the submitted version.

## Competing interests

The authors declare no competing interests.
