## [Peer Review File · Nature Communications]

Household transmission of SARS-CoV-2 Omicron variant of concern subvariants BA.1 and BA.2 in DenmarkEditorial Note: This manuscript has been previously reviewed at another journal that is not operating a transparent peer review scheme. This document only contains reviewer comments and rebuttal letters for versions considered at *Nature Communications* .

REVIEWERS' COMMENTS

Reviewer #1 (Remarks to the Author):

Authors have addressed previous comments. The availability of additional data on variants is valuable. I just had one suggestion based on the last sentence of the Results section

Authors have information that 2%-7% of contacts were infected with a different variant to an index case. This does not mean that 93%-98% of infected contacts were infected by their index case. But this information can be used in a statistical model to estimate the proportion of infected contacts that were infected by the index case. A rough guess from your data would be 90%, because some of the contacts with same lineage as index case would have been infected outside the home. Analysis of this type reported here for example <https://pubmed.ncbi.nlm.nih.gov/21878814/>

Response to Reviewers

Round 2

Reviewer #1 (Remarks to the Author):

R1 Authors have addressed previous comments. The availability of additional data on variants is valuable. I just had one suggestion based on the last sentence of the Results section

Authors have information that 2%-7% of contacts were infected with a different variant to an index case. This does not mean that 93%-98% of infected contacts were infected by their index case. But this information can be used in a statistical model to estimate the proportion of infected contacts that were infected by the index case. A rough guess from your data would be 90%, because some of the contacts with same lineage as index case would have been infected outside the home. Analysis of this type reported here for example <https://pubmed.ncbi.nlm.nih.gov/21878814/>

AU We agree with the reviewer in the sense that this measure is not a final proof of transmission. It is however a necessary condition, as we also write in the appendix [S4.2 *Misclassification of cases; iii) Misclassification of community cases as secondary household cases*].

We have now changed the sentence from “We found that households located in municipalities with predominantly other variants have a slightly lower intra-household correlation of variants of 93% (specification IV).” to “We found that in households located in municipalities with predominantly other variants, secondary cases have a slightly lower probability of being infected with the same variant as the primary case (93%, specification IV).” (appendix page 21-22).